# UV-free Texture Generation with Denoising and Geodesic Heat Diffusions

**Simone Foti**     **Stefanos Zafeiriou**     **Tolga Birdal**

Department of Computing
Imperial College London

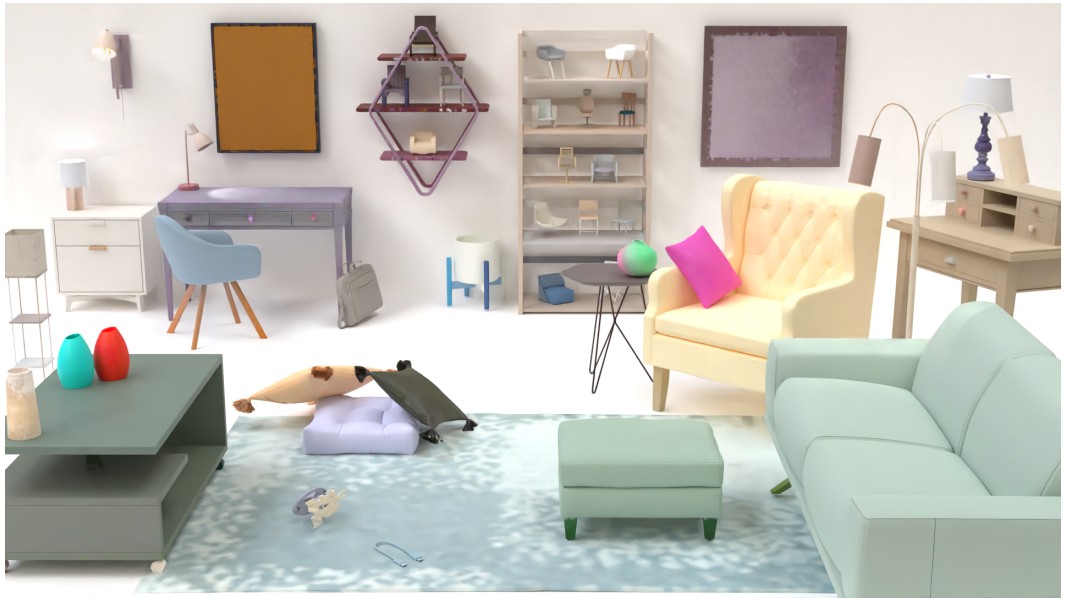

Figure 1: Random textures generated by our method, *Uv3*-TeD, on the surface of general objects from the Amazon Berkeley Object dataset and of chairs from ShapeNet (miniatures on the shelves).

## Abstract

Seams, distortions, wasted UVspace, vertex-duplication, and varying resolution over the surface are the most prominent issues of the standard UV-based texturing of meshes. These issues are particularly acute when automatic UV-unwrapping techniques are used. For this reason, instead of generating textures in automatically generated UV-planes like most state-of-the-art methods, we propose to represent textures as coloured point-clouds whose colours are generated by a denoising diffusion probabilistic model constrained to operate on the surface of 3D objects. Our sampling and resolution agnostic generative model heavily relies on heat diffusion over the surface of the meshes for spatial communication between points. To enable processing of arbitrarily sampled point-cloud textures and ensure long-distance texture consistency we introduce a fast re-sampling of the mesh spectral properties used during the heat diffusion and introduce a novel heat-diffusion-based self-attention mechanism. Our code and pre-trained models are available at github.com/simofoti/UV3-TeD.

38th Conference on Neural Information Processing Systems (NeurIPS 2024).

# 1 Introduction

Meshes are surface discretisations intentionally designed to represent the geometry of 3D objects. Since real objects are more than just their geometry, meshes are frequently augmented with textures, representing appearances. Currently, textures are mostly represented as images that can be wrapped on the mesh via a UV-*mapping* procedure that maps every point on the surface of the mesh into a point on the UV-image-plane where texture information are stored. Textures are therefore mostly generated on images, but considering that UV-maps intrinsically suffer from distortions, seams, wasted UV-space, vertex-duplication, and varying resolution [19], is UV-mapping really the best approach? In this work, we wonder: what if instead of generating a texture on a plane and then wrapping it onto a shape we could directly generate a texture on the curved surface of the object?

The capability of generating textures by avoiding UV-unwrapping and mapping altogether alleviates the post-processing issues caused by the UV-mapping and can save time and resources for many 3D artists, while generally improving the realism of the textures by avoiding distortions, seams, or stretching artefacts. In addition, it could enable the creation of priors for a variety of computer vision tasks ranging from shape and appearance reconstruction to object detection, identification, and tracking. More generally, a texture representation adhering to the actual geometry of the surface could result in smaller memory footprints without compromising on the rendering quality.

While most state-of-the-art methods focus on generating textures in UV-space [23, 7, 62, 56, 13, 34, 12, 42, 41] and thus inherit all the drawbacks of UV-mapping (Fig. 2, *left*), we propose to represent textures with unstructured point-clouds sampled on the surface of an object and devise a technique to render them without going through UV-mapping (Fig. 2, *right*). We generate point-cloud textures with a denoising diffusion probabilistic model operating exclusively on the surface of the meshes. This is fundamentally different from generating coloured point-clouds [58] or triplane-based implicit textures [8]: while our work respects the geodesic information provided by the meshes, they operate in the Euclidean space and ignore the topology of the objects they seek to texturise. When compared to methods like [49], which can generate textures directly on the surface, our method has the advantage of not requiring any remeshing operation and being adaptable to different sampling resolutions. In addition, unlike many other texture generation methods [7, 12, 42, 49], we generate albedo textures that by not factoring in the environment can be rendered with different lighting conditions to achieve more photorealistic results (Fig. 1). Finally, while most methods are class-specific, our method can be trained on datasets containing objects of different classes (Fig. 1). Our key contributions are:

1. We create a denoising diffusion probabilistic model generating point-cloud textures by operating only on the surface of the meshes,
2. We introduce a novel attention layer based on heat diffusion and farthest point sampling to improve the recently proposed DiffusionNet blocks [48] by facilitating global communication across the entire surface of the object,
3. We propose a mixed Laplacian operator to ensure that heat diffusion can spread even in the presence of topological errors and disconnected components while still mostly relying on the provided topological information,
4. We devise an online sampling strategy that allows us to sample point-clouds and their spectral properties during training without requiring to recompute them from scratch.

# 2 Related Work

**Texture Representations and Rendering**. Textures are traditionally represented by images which are mapped onto 3D shapes via a UV-mapping. Since manual UV-mapping is complex and labour-intensive, many methods tried to perform it automatically while attempting to address some of the most common artefacts: seams and distortion [44, 39]. Other texture representations such as [19, 50, 2, 61] have been proposed without finding a level of adoption comparable to the one of UV-textures, which are still the de facto standard for modelling the appearance of meshes. Although arguably the most convenient shape representation for computer graphics applications, meshes are not the only data structure used to represent 3D shapes. In fact, point-clouds are equally spread and they can also be associated with appearance information, stored as colour values associated to each point. Many techniques have tried to reduce their sparsity while rendering [46, 45, 10], but when photo-realism is required, textured meshes are still a superior representation which can better approximate continuous surfaces. Given the strengths and limitations of both representations, we propose to adopt a hybrid approach where the geometry is represented by a mesh and its appearance

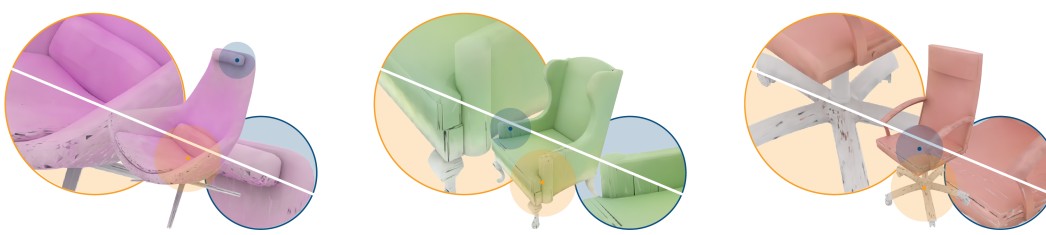

Figure 2: Qualitative comparison between point-cloud-textures (*top-right* halves) and automatically wrapped UV-textures (*bottom-left* halves). All textures were generated by Point-UV Diffusion in order to showcase some of the most common issues of UV-mapping. Although the method generated good quality textures as point-clouds, projecting them in UV-space introduces significant artefacts.

by an unstructured point-cloud texture. This can prevent seams, distortions, unused UV-space, and varying resolution while still enabling the rendering of continuous surfaces. Hybrid approaches mixing meshes and point-clouds have also been proposed by [18] and [60, 59]. However, in [18] the appearance was present in both representations, which were both rendered, and [60, 59] used highly structured point-clouds with points regularly positioned on the mesh faces. The representation of [60, 59] can be used only for high-resolution textures requiring significantly more points than the number of vertices, thus making it incompatible with current geometric deep learning models. For this reason, we use unstructured point-cloud textures arbitrarily sampled at any required resolution.

**Texture Generation**. Many texture generation techniques have been proposed, each relying on a different representation. Considering the wide adoption of UV-textures, and the maturity of deep-learning techniques operating on the image domain, it is not surprising that most methods still rely on UV-mapping and generate UV-images to wrap on meshes [23, 7, 62, 56, 13], or simultaneously optimise meshes and texture to achieve the desired result [34]. Alternatively, another common approach consists in using a generative model to generate depth-conditioned images from multiple viewpoints. These images are then projected onto a mesh, refined, and stored as UV-textures [12, 42, 41]. Other methods try to map textures on UV-spheres [14], tri-planes [22, 53], NeRFs [1, 32], or implicit functions [35], but they either fail to operate on the real geometry of the object or they end up-projecting the textures on a UV-plane. Even Point-UV Diffusion [58], whose coarse stage is conceptually similar to ours because it generates coloured point-clouds using point-cloud operators, effectively projects points in UV-textures that are refined with image diffusion models. Unfortunately, especially at the coarse stage, this often results in the visible artefacts that are typical of this parametrisation (Fig. 2). A method capable of operating directly on the surface of the objects is Texturify [49], which shows remarkable results adopting a shape-conditioned Style-GAN convolving coloured quad-faces. The main limitation of this method is its need to uniformly re-mesh the input shapes to a fixed resolution, potentially affecting the quality of the original mesh. Most of these methods are trained on class-specific shapes suggesting their difficulties in dealing with multi-class datasets. While our method can actually operate on datasets with shapes belonging to different classes, some methods exacerbate the single-class limitation, focusing exclusively on training their models on single shapes to then generate texture variations [54, 33]. Similarly, manifold diffusion fields [20] are capable of generating continuous functions –such as textures– over Riemannian manifolds. Unfortunately, they generate functions only on manifold meshes and they are always trained on single-shape datasets with multiple function variations. It is unclear whether their method would be capable of generalising to different geometries.

## 3   Notation and Background

We define a mesh as $\mathcal{M} = \{\mathbf{V}, \mathbf{F}\}$, with $\mathbf{V} \in \mathbb{R}^{V \times 3}$ representing the positions of the $V$ vertices sampled on the surface ($\mathcal{S}$) of a shape, and $\mathbf{F} \in \mathbb{N}^{F \times 3}$ the set of triangular faces describing the connectivity of the vertices. Throughout this work, we assume that the vertex positions and the mesh topology are given, but different for every shape we want to texturise. We think of textuers as continuous functions $x : \mathcal{S} \to \mathcal{X}$ mapping points on $\mathcal{S}$ to a signal space $\mathcal{X}$ that, without any loss of generality, corresponds to the albedo colours, *i.e.*, $\mathcal{X} = \mathbb{R}^3$. In practice, we operate on textures defined as coloured point-clouds with colours $\mathbf{X} = x(\mathbf{P}) \in \mathbb{R}^{P \times 3}$, where $\mathbf{P} \in \mathbb{R}^{P \times 3}$ represents the $P$ 3D coordinates of the points sampled on $\mathcal{S}$. Note that, in general, we assume $\mathbf{P} \neq \mathbf{V}$ (and $P \neq V$) as the texture point-clouds do not necessarily need to follow the same vertex distribution.

Before introducing the main contributions of our work in Sec. 4, we formalise two pillars on which we base our method: the denoising diffusion models and the Laplace Beltrami operator.

**Denoinsing Diffusion Probabilistic Model**. Denoising Diffusion Probabilistic Models [26] (DDPMs) are now a well-established class of generative models that rapidly found adoption across different fields[11, 31]. They are parameterised by a Markov chain trained using variational inference and are essentially characterised by three steps: a forward noising procedure, a backward denoising, and a sampling procedure that is used during inference. During the forward process, a training sample $\mathbf{X}_0 \sim p(\mathbf{X}_0)$ corresponding to a point-cloud texture and coming from the original textures distribution at timestep 0 is iteratively perturbed to $\{\mathbf{X}_t\}_{t=1}^T$ by progressively adding a small amount of isotropic Gaussian noise $\epsilon_0 \sim \mathcal{N}(\mathbf{0}, \mathbf{I})$. Being $p(\mathbf{X}_t|\mathbf{X}_{t-1}) := \mathcal{N}(\mathbf{X}_t; \sqrt{1 - \beta_t}\mathbf{X}_t, \beta_t\mathbf{I})$ a single step in the discrete forward chain with noise schedule $\beta_t$, we represent the full chain as $p(\mathbf{X}_T|\mathbf{X}_0) = \prod_{t=1}^T p(\mathbf{X}_t|\mathbf{X}_{t-1})$. Similarly, a generic step in the chain can be obtained as $p(\mathbf{X}_t|\mathbf{X}_0) := \mathcal{N}(\mathbf{X}_t; \sqrt{\bar{\alpha}_t}\mathbf{X}_t, (1 - \bar{\alpha}_t)\mathbf{I})$, where $\mathbf{I}$ is the identity matrix, $\bar{\alpha}_t = \prod_{s=1}^t \alpha_s$, and $\alpha_t = 1 - \beta_t$. This implies that $\mathbf{X}_t = \sqrt{\bar{\alpha}_t}\mathbf{X}_0 + \sqrt{1 - \bar{\alpha}_t}\epsilon_0$. In DDPM models like ours, the reverse process is parameterised by a neural network trained to predict the noise that needs to be progressively removed. This process is formulated as $p_\theta(\mathbf{X}_{t-1}|\mathbf{X}_t) := \mathcal{N}\left(\mathbf{X}_{t-1}; \frac{1}{\sqrt{\alpha_t}}\left(\mathbf{X}_t - \frac{1-\alpha_t}{\sqrt{1-\bar{\alpha}_t}}\epsilon_\theta(\mathbf{X}_t, t)\right), \beta_t\mathbf{I}\right)$, where the variance is empirically fixed and the mean is leveraging the noise prediction network $\epsilon_\theta(\mathbf{X}_t, t)$. The variational inference objective can thus be simplified to $\mathcal{L} = \mathbb{E}_{\mathbf{X}_t, t}\left[\|\epsilon_t - \epsilon_\theta(\mathbf{X}_t, t)\|_2^2\right]$. Finally, the sampling process follows the reverse process where the trained network transforms noise samples coming from the terminal distribution $\mathbf{X}_T \sim p(\mathbf{X}_T)$ into the denoised $\hat{\mathbf{X}}_0 \sim p_\theta(\mathbf{X}_0) \approx p(\mathbf{X}_0)$.

**Eigenproperties of Laplace Beltrami Operator**. The Laplace Beltrami operator (LBO) plays an essential role in geometry processing. For triangle meshes, the LBO is usually based on a cotangent formulation [38, 63] derived from finite element analysis. The cotangent Laplacian $\mathbf{L} \in \mathbb{R}^{V \times V}$ is a sparse matrix with elements proportional to the cotangent of the angles subtended by the edges, and it is associated to a diagonal mass matrix $\mathbf{M} \in \mathbb{R}^{V \times V}$ whose diagonal elements are proportional to the total area of the faces surrounding each vertex [47]. The eigendecomposition of the LBO, $\mathbf{L}\mathbf{\Phi} = \mathbf{\Lambda}\mathbf{M}\mathbf{\Phi}$, determines a set of orthonormal eigenvectors $\mathbf{\Phi} := [\phi_k]_{k=1}^K \in \mathbb{R}^{V \times K}$ corresponding to the $K$ smallest eigenvalues $\mathbf{\Lambda} := [\lambda_k]_{k=1}^K \in \mathbb{R}^K$ of the weak Laplacian and its mass matrix. These eigenvalues and eigenvectors have been intensively studied in spectral geometry because not only can they be used as global and local shape descriptors, but they can also be used to formulate surface operations such as heat diffusion [63]. As the name suggests, heat diffusion regulates the physical heat dispersion. This phenomenon can be modelled on any discrete surface representation with a Laplacian operator and it is resolution, sampling, and representation independent.

# 4 UV-free Texture Diffusion (UV3-TeD)

We now describe UV3-TeD, our generative model for learning point-cloud textures built upon heat-diffusion-based operators specifically designed to operate on the surface of the input shapes (detailed in Sec. 4.1 and depicted in Fig. 4). Our diffusion model UV3-TeD operates on a noised version of the colours, $\mathbf{X}_t$, and predicts a denoised $\mathbf{X}_{t-1}$ through a U-Net [43] shaped architecture (Fig. 9), backed by novel *attention-enhanced heat diffusion blocks* (Sec. 4.1). We represent every mesh, $\mathcal{M}$, by a novel mixed LBO, informed both by the geometry and the topology of $\mathcal{M}$(Sec. 4.2). We further introduce an *online sampling*, in order to obtain a point-cloud $\mathbf{P}$ with corresponding albedo colours $\mathbf{X}$ and tailored spectral operators (Sec. 4.3). UV3-TeD is trained with a denoising objective described in Sec. 3, using heterogeneous batching. Meshes with a point-cloud texture can finally be rendered by the nearest-neighbour interpolation we detail in Sec. 4.4.

## 4.1 Attention-enhanced Heat Diffusion Blocks

**Diffusion Blocks (DB)**. Our blocks are inspired by DiffusionNet [48], consisting of three separate learnable parts: heat diffusion, spatial gradient features, and a vertex-wise multi-layer perceptron (MLP). The heat diffusion process is used to disperse and aggregate information on a surface and it has a closed-form solution leveraging the spectral properties of the LBO. Since we aim to operate on point-cloud textures while leveraging topological and geometric information provided by the mesh, we use our tailored versions of $\mathbf{M}$ and $\mathbf{\Phi}$ later described in Sec. 4.3 and named $\mathbf{\Phi}_p$ and $M_p$

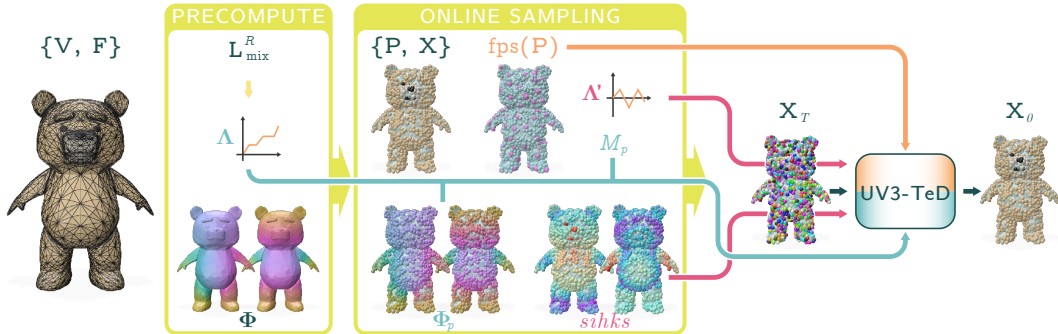

Figure 3: Framework of UV3-TeD. Given a mesh $\mathcal{M} = \{\mathbf{V}, \mathbf{F}\}$ we precompute the proposed mixed Laplacian ($\mathbf{L}_{\text{mix}}^R$) and its eigendecomposition ($\mathbf{\Lambda}$ and $\mathbf{\Phi}$). During the online sampling we compute a coloured point-cloud $\{\mathbf{P}, \mathbf{X}\}$ alongside its spectral quantities and other information used by our network (Fig. 9). In particular, eigenvalues $\mathbf{\Lambda}$, sampled eigenvectors $\mathbf{\Phi}_p$, and approximate mass $M_p$ are used to compute the heat diffusion operations (Eq. (1)); the farthest point samples $\text{fps}(\mathbf{P})$ are used in the proposed diffused farthest-sampled attention layers (Fig. 4), and the scale invariant heat kernel signatures $sihks$ and slope-adjusted eigenvalues $\mathbf{\Lambda}'$ are used as shape conditioning. UV3-TeD leverages these information to generate coloured point-clouds ($\mathbf{X}_0$) from noise ($\mathbf{X}_T$).

respectively. Being $\mathbf{Y}_p \in \mathbb{R}^{P \times Y}$ a generic field defined on $\mathbf{P}$, the heat diffusion layer is defined as:

$$\text{diffuse}(\mathbf{Y}_p, h) := \mathbf{\Phi}_p \begin{bmatrix} e^{-\lambda_1 h} \\ \vdots \\ e^{-\lambda_K h} \end{bmatrix} \odot (\mathbf{\Phi}_p^{\text{T}} M_p \mathbf{Y}), \tag{1}$$

where $\odot$ is the Hadamard product, T is the transpose operator, and $h$ is the channel-specific learnable parameter indicating the heat diffusion time and effectively regulating the spatial support of the operator, which can range from local ($h \to 0$) to global ($h \to +\infty$). Since this block supports only radially symmetric filters about each point, it is combined with a spatial gradients features block that expands the space of possible filters by computing the inner product between pairs of feature gradients undergoing a learnable per-vertex rotation and scaling transformation in the tangent bundle (see [48] for more details). The spatial gradients are computed online on the sampled point-cloud with our faster implementation (see App. A.1). Then, the input is concatenated with the output of these two blocks and passed to a per-vertex MLP (see Fig. 4, *bottom*).

We further add a time embedding representing the denoising step of the DDPM and introduce a group normalisation [55] to stabilise training after the time injection. We also add a linear layer on the skip connection when input and output channels differ., *e.g.*, when skip connections from the downstream branch of the U-Net are concatenated with the features of the upstream branch.

**Enhancing DB via Farthest-Sampled Attention**. As depicted in Fig. 4, each of our Attention-enhanced Heat Diffusion Blocks concatenate three Diffusion Blocks and combine them with our diffused farthest-sampled attention layer. Even though with the Diffusion Blocks alone it is theoretically possible to achieve global support when $h \to +\infty$, the longer the heat diffusion time, the closer the diffused features become to the average of the input over the domain. This can result in less meaningful features causing texture inconsistencies between distant regions. To improve long-range consistency we introduce attention layers in each network's block alongside the other operators. Since directly performing the scaled dot product operation characterising attention modules on full-resolution point-cloud textures would be prohibitive, we build upon the heat diffusion concept and define a more efficient attention operator (Fig. 4, *top*).

We start by heat-diffusing the $\mathbf{Y}_p^{(i-1)}$ features predicted by the previous layers over $\mathbf{P}$ to spread information across the surface geodesically. Then, we collect the spread information (i.e., $\text{diffuse}(\mathbf{Y}_p^{(i-1)}, h)$) on a subset of the diffused features, $\mathbf{S} \in \mathbb{R}^{S \times C}$, which is obtained by selecting the diffused features with $C$ channels corresponding to the $S$ farthest samples [40] of $\mathbf{P}$. $\mathbf{S}$ is then fed to a multi-headed self-attention layer [51], where a set of linear layers first computes queries, keys, and values for each head, then computes a scaled dot-product attention, concatenates the results across the different heads and, after going through an additional linear layer, produces a new set of features over the farthest samples. These features still reside on the farthest samples. To spread them

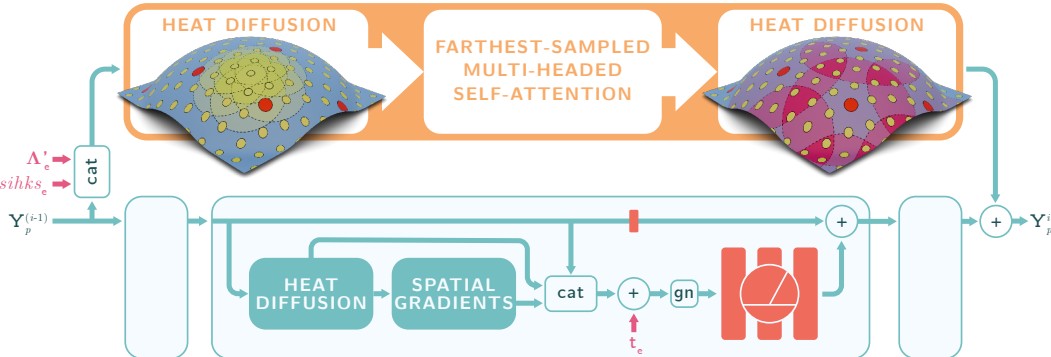

Figure 4: Attention-enhanced Heat Diffusion block. Three consecutive Diffusion blocks (*bottom*) inspired by [48] and conditioned with a denoising time embedding are combined with a diffused farthest-sampled attention layer (*top*). The proposed attention, conditioned with local and global shape embeddings ($sihks_e$ and $\mathbf{\Lambda}'_e$), first spreads information to all the points on the surface, before computing a multi-headed self-attention on the features of the farthest samples (red points), and finally spreads them back to all the points with another heat diffusion.

across the entire surface, we set the features of the other points to zero and perform another heat diffusion. The output of the diffused farthest-sampled attention is re-combined with the output of the other blocks learning a per-channel weighting constant.

**Conditioning**. While other point-cloud networks require additional inputs to represent the positions of the input points alongside their features, we just provide noised colours as inputs because the diffusion process intrinsically operates on the surface of the shapes we want to texturise. Nevertheless, we do provide geometric and positional conditioning to the diffused farthest-sampled attention layers, which are otherwise unaware of the relative position of their inputs. Instead of using $\mathbf{P}$ to compute the positional conditioning directly, we rely on the scale-invariant heat kernel signatures ($sihks$) [6], intrinsic local shape descriptors that are not only sampling and format agnostic but also isometry and scale-invariant. The geometry conditioning is obtained from the eigenvalues $\mathbf{\Lambda}$, which, like in [24], are normalised by $\text{Area}(\mathcal{M})$ and deprived of their slope as:

$$\mathbf{\Lambda}' = \left\{ \lambda'_k \ \middle| \ \lambda'_k = \frac{\lambda_k}{\text{Area}(\mathcal{M})} - 4\pi * k, \quad \text{for } k = 1, \dots, K \right\}. \tag{2}$$

Eigenvalues processed as in Eq. (2) can still be used as global shape descriptors that besides having the advantage of being scale invariant also fluctuate over a straight line, becoming easier to process for a neural network. Intuitively, $sihks$ tell us the intrinsic coordinates of a point, while $\mathbf{\Lambda}'$ whether we are supposed to generate a texture on a chair, a sofa, a vase, or something else. Both are embedded with a MLP and the resulting geometry embeddings are concatenated with the point features.

### 4.2   Mixed Robust Laplacian

To operate on real-world datasets we propose a mixed Laplacian operator which is robust to any triangle mesh and can better diffuse heat in the presence of complex topological structures (see Fig. 5, *left*). Our mixed robust LBO ($\mathbf{L}^R_{\text{mix}}$) is defined as:

$$\mathbf{L}^R_{\text{mix}} = (1 - \varrho)\mathbf{L}^R_m + \varrho\mathbf{L}^R_p, \qquad \text{with } \varrho \in \mathbb{R}. \tag{3}$$

Instead of using the $cotan$-LBO directly, we use the robust mesh Laplacian $\mathbf{L}^R_m$ [47], computed on the vertices of the mesh, as it provides robustness to non-orientable and non-manifold meshes. $\mathbf{L}^R_m$ ensures that heat is geodesically diffused, while $\mathbf{L}^R_p$, its point-cloud counterpart, enables communication between distinct or disconnected components of a mesh. A small $\varrho$ value leads to diffusing heat on the surface while allowing for some heat transmission to neighbouring regions (see Fig. 5, *right*).

### 4.3   Online Sampling of Points, Colours, and Spectral Operators

Our sampling strategy is at the core of our method as it provides an efficient sampling strategy that can be used online without hindering training speeds. In particular, Poisson Disk Sampling produces a point-cloud with regularly-spaced points, enabling us to approximate the mass matrix quickly.

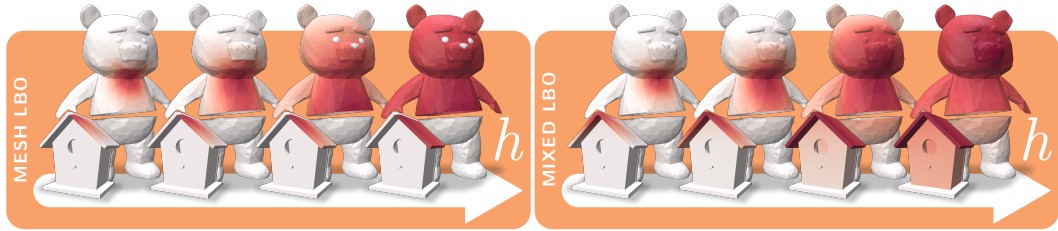

Figure 5: Heat diffusion on Ted sliced on the belly and on a topologically disconnected birdhouse. Using the mesh LBO prevents heat from spreading to disconnected regions, this is particularly visible on Ted as heat does not spread over the nose, mouth, and legs. Similarly, on the birdhouse heat spreads only on the right-hand side of the roof. Using our mixed LBO formulation heat can spread over the entire shape even in the presence of topological errors and disconnected components.

To avoid recomputing the eigendecomposition of our Laplacian operator ($\mathbf{L}^R_{\mathrm{mix}}$) on the sampled point-cloud, we recycle the spectral operators precomputed on the vertices of the meshes. Finally, we describe how colours are sampled during training.

**Poisson Disk Sampling** (PSD) [5]. PDS produces a point-cloud $\mathbf{P} \in \mathbb{R}^{P \times 3}$, by uniformly sampling points on the surface of a mesh. This is achieved with a parallel dart-throwing algorithm that uses a uniform radius $r$ across the surface. The samples $\mathbf{p}_i \in \mathbf{P}$ are randomly distributed on the surface but remain a minimum distance of $r$ away from each other. Since PDS is designed to operate given a radius rather than a desired number of points $P^*$, the radius can be estimated from the ideal quality measure expected from the radius statistics introduced in [5] (see also [4]) :

$$\rho = \frac{r}{2}\Big(\frac{2\sqrt{3}P^*}{\mathrm{Area}(\mathcal{M})}\Big)^{\frac{1}{2}} \approx 0.7 \tag{4}$$

**Mass Matrix**. Since the point-cloud textures have been sampled using PDS, we hypothesise that the distance between neighbouring points will equal the radius $r$ used by PDS. A triangulation of such points would result in equilateral faces with area $\mathrm{Area}(\mathbf{F}_{ijk}) = \frac{r^2}{2}\sin(\frac{\pi}{3}) = \frac{r^2\sqrt{3}}{4}$. Therefore, said $Q$ the number of faces incident to each vertex $\mathbf{p}_i$ and computing the radius with Eq. (4) we can approximate the mass matrix as:

$$\mathbf{M}_{ii} = \frac{1}{3}\sum_{ijk \in \mathbf{F}} \mathrm{Area}(\mathbf{F}_{ijk}) \approx \frac{Q}{3}\frac{\sqrt{3}}{4}r^2 \approx \frac{(0.7)^2 Q}{6P^*}\mathrm{Area}(\mathcal{M}), \ \forall i. \tag{5}$$

Since we never explicitly compute a triangulation of the point-cloud texture, we estimate $Q$ on $\mathcal{M}$. Also, considering that the mass matrix derived in Eq. (5) has the same value on the diagonal elements, we represent it with a scalar, $M_p$.

**Eigenvalues and Eigenvectors**. The eigenvalues of LBO are considered global shape descriptors, as such, they are sampling-independent. Eigenvectors are on the other hand defined on the vertices of the mesh on which LBO was computed. However, as mentioned in Sec. 3, a mesh $\mathcal{M}$ is effectively discretising a continuous surface $\mathcal{S}$. Similarly, a signal on the vertices of the mesh can be thought of as a discretisation of the continuous function defined on $\mathcal{S}$. For this reason, eigenvectors can be resampled by interpolating the values of the eigenvectors defined at the vertices of the mesh. We indicate these with $\mathbf{\Phi}_p \in \mathbb{R}^{P \times K}$.

**Colour Sampling**. Although we advocate for a new texture representation based on point-clouds, the most widely adopted representation is still based on UV-mapping. Hence, for every point sampled with PDS, we also query the colour stored in the UV image plane at its corresponding UV coordinates. When UV textures are not provided, we sample the base colour instead. Following this procedure, we obtain coloured point-clouds $\{\mathbf{P}, \mathbf{X}\}$ that we use as point-cloud textures during training.

When images have a significantly higher resolution than the desired point-cloud texture resolution, we resize the image texture before sampling. We assume that properly textured meshes should intentionally have big UV triangles where a high texture resolution is required. Being $\triangle^{uv}$ the $N$ biggest triangles in UV space and $\triangle^{3D}$ the corresponding triangles on the mesh, to estimate the scaling factor ($s$) needed to obtain the desired image texture size, we compute the square root of the

ratio between the number of samples on $\triangle^{3D}$ and the number of pixels in $\triangle^{uv}$:

$$s = \left[ \frac{\frac{1}{N} \sum_{n=1}^{N} \text{Area}(\triangle^{3D}) / \text{Area}(\mathbf{F}_{ijk})}{(W \times H)\left(\frac{1}{N} \sum_{n=1}^{N} \text{Area}(\triangle^{uv})\right)/1^2} \right]^{\frac{1}{2}} . \tag{6}$$

The number of samples in $\triangle^{3D}$ is estimated dividing their average area by the approximate area of the PD sampled point-cloud texture. The number of pixels in $\triangle^{uv}$ by estimating the fraction of UVspace occupied by the biggest triangles and multiplying it by the number of pixels in the image plane, which is computed as the product between image width and height ($W \times H$). In practice, we set $N = 250$ to consider a significant number of triangles and use $3s$ instead of $s$ to account for non-perfectly textured meshes which retain useful high-resolution content in small UVtriangles.

### 4.4 Rendering Point-Cloud Textures

We rely on Mitsuba3 [27], a physically-based differentiable renderer, and implement a new class of textures: the point-cloud textures that we previously characterised with the $\{\mathbf{P}, \mathbf{X}\}$ pair. When a ray intersection occurs and the point-cloud texture is queried, we compute the three nearest point-cloud neighbours to the hit point and interpolate their colour values. This is analogous to the standard texture querying that would occur in UVspace. Note that the nearest neighbours are computed using Euclidean rather than geodesic distances. When enough points are sampled, this is a reasonable assumption that keeps rendering times low.

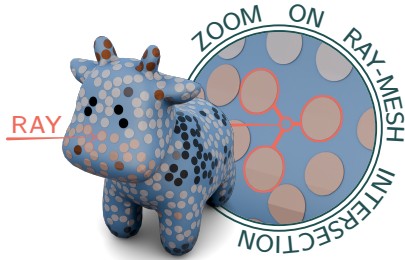

Figure 6: Rendering a point-cloud textured cow [16]. When a ray intersects the mesh, we interpolate the colours of the three nearest texture points.

## 5 Experiments

**Datasets**. We conduct experiments on two datasets, the chairs of ShapeNet [9] and the Amazon Berkeley Objects (ABO) dataset [15]. The chair category of ShapeNet has often been used for texture generation on 3D shapes because, compared to the other categories, it has a high number of samples with relatively high texture resolutions. This motivated us to train UV3-TeD on these data. However, driven by the objective of building a model capable of operating across multiple categories, we also decided to leverage the less widely used ABO. Despite its more limited adoption, this dataset contains multiple object categories with good-quality meshes and textures.

Since with UV3-TeD mesh and texture resolution are independent, we pre-filter data with more than $60,000$ vertices. This choice doesn't hinder the quality of the generated point-cloud textures but reduces the GPU memory consumption during training. As we are interested in generating textures, we also discard meshes that have coloured parts, but no textures. In both cases, we operate a $90 : 5 : 5$ split between train, test, and validation sets. The filtering and data split leave us with $4,633$ chairs for training, $266$ for validation, and $317$ for testing. On ABO we have $6,476$ shapes for training, $364$ for validation, and $443$ for testing.

**Implementation Details**. We implement our method using PyTorch [37], Pytorch Geometric [21] and Diffusers [52]. We use 32 $sihks$, $K = 128$ eigenvalues, and a mixed-LBO weighting of $\varrho = 0.05$. We train our models using the AdamW [30] optimiser for $400$ epochs on chairs and $250$ on ABO, with a learning rate of $1e^{-4}$ and a cosine annealing with $500$ warmup iteration steps. We use $T = 1,000$ DDPM timesteps, $S = 250$ farthest point samples in the attention layers, and $P^* = 5,000$ target PDS samples. Since $P^*$ is used to estimate the mesh-specific PDS radius $r$, our point-cloud textures often have a slightly different amount of points $P$. Thus, we made all our layers suitable for heterogeneous batching. Our batch size is set to $8$ on ShapeNet chairs and to $6$ on ABO.

**Unconditional Texture Generation**. In Fig. 7 *top* we showcase our texture generation results on chairs from ShapeNet, in Fig. 8 we depict results on objects from ABO, and in Fig. 1 we combine textured object from both datasets rendered with more advanced lighting (more textured samples are provided in App. A.2). Then, we proceed to compare our method against the state-of-the-art.

Although we acknowledge that many state-of-the-art methods operating in UV-space have generated impressive results, we want to highlight that they operate on images, a data type which has been vastly explored by the Deep Learning community and where models are well-engineered, mature in

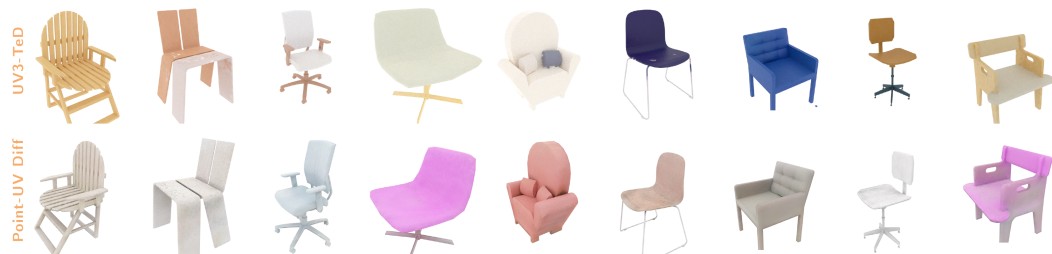

Figure 7: Qualitative comparison between PointUVDiff (pcl-texture) and *UV3*-TeD (ours). Our textures are more diverse and more semantically meaningful. All shapes belonged to the test set.

terms of efficiency and quality, and trained on larger datasets. We find it important to note that we do not aim to compete against UV techniques, but rather attempt to prompt a paradigm shift towards a direction that will not require UV-mapping, with its many unsolved issues. As detailed in Sec. 2, we consider the coarse stage of Point-UV Diffusion [58] and Texturify [49] to be the best non-UV-based method currently available as well as the most relevant works to ours. The former already extensively proved its superiority over Texturify and [35] on the chairs of ShapeNet. On the same data, Texturify made additional comparisons against UV ([57] and a UV Baseline), Implicit [35], Tri-plane [8], and sparse grid [17] methods, outperforming all of them across all metrics. For this reason, we focus our comparison on Point-UV Diffusion, which does not have shadows baked in the texture and is therefore better suited to be rendered with our pipeline (Sec. 4.4). Because we adopt physically-based rendering techniques rather than rasterisation, we re-compute FID and KID scores [25, 36, 3] for Point-UV Diffusion. Each shape is rendered from 5 random viewpoints with the camera pointing at the object at an azimuth angle sampled from $\mathcal{U}[0, 2\pi]$ and an elevation sampled from $\mathcal{U}[0, \pi/3]$. We also compute the LPIPS [64] metric, measuring the diversity of the generated textures. For this, we generate 3 texture variations for each shape, render them, and compute the LPIPS values for all the possible pairs of images with the same underlying shape. The results are then averaged for each method. We also evaluate two different scenarios: one in which we emulate the original formulation and render shapes whose point-cloud textures have been projected in UV-space, and one where we do not project their point-clouds in UV-space using our rendering technique instead. As we can observe from the quantitative results reported in Tab. 1, our method significantly outperforms Point-UV Diffusion across all metrics. In addition, as we can observe from the samples reported in Fig. 1 and Fig. 7, our method can generate more diverse samples, which are also more capable of capturing the semantics of the different object parts. Interestingly, this is achieved without providing any semantic segmentation. UV3-TeD also significantly outperforms a DiffusionNet DDPM in sample quality while maintaining comparable diversity (Tab. 1). This baseline mimics UV3-TeD by leveraging a DDPM model with a U-Net-like architecture having as many layers as ours, but without shape conditioning and using the point operators of [48]. Therefore, not only there were no farthest-sampled attention layers, but no online sampling strategy was used and the point-cloud Laplacian and mass matrix were used instead. Colours were still sampled like for UV3-TeD. All the hyperparameters matched ours.

Table 1: Quantitative comparison on the chairs of ShapeNet.

| Method | FID ($\downarrow$) | KID $\times 10^{-4}$ ($\downarrow$) | LPIPS ($\uparrow$) |
|---|---|---|---|
| PointUVDiff [58] (uv) | 63.35 | 83.19 | 0.08 |
| PointUVDiff [58] (pcl-texture) | 65.09 | 126.18 | 0.09 |
| DiffusionNet [48] DDPM | 116.58 | 468.09 | **0.24** |
| UV3-TeD (ours) | **54.20** | **42.17** | 0.21 |

**Ablation Studies**. We here perform multiple ablations to examine how much each model component, conditioning, and choice contributes to the overall performance. The ablations are performed training the model for 50 epochs on the ABO (multi-class) dataset. Ablations are quantitatively evaluated on 100 test shapes using the three main metrics previously used during the comparisons: the FID, KID, and LPIPS scores. Results are reported in Tab. 2. Overall, our final model (UV3-TeD) reports the best quality scores while maintaining good diversity scores. A detailed discussion of the different ablations is provided in App. A.2.

Table 2: Ablation studies on our UV3-TeD on ABO. Models were trained for 50 epochs.

| | FID ($\downarrow$) | KID $\times 10^{-4}$ ($\downarrow$) | LPIPS ($\uparrow$) |
|---|---|---|---|
| UV3-TeD (ours) | **77.14** | **58.59** | 0.14 |
| w/o farthest-sampled attention | 83.16 | 103.42 | 0.10 |
| w/o $\mathbf{\Lambda}'$ | 79.96 | 66.29 | 0.15 |
| w/o $sihks$ | 78.46 | 70.90 | 0.14 |
| $\mathbf{\Phi}_p$ instead of $sihks$ | 79.62 | 65.17 | 0.14 |
| $\varrho = 0$ ($\mathbf{L}_m^R$ instead of $\mathbf{L}_{\text{mix}}^R$) | 78.07 | 62.64 | 0.15 |
| $\varrho = 1$ ($\mathbf{L}_p^R$ instead of $\mathbf{L}_{\text{mix}}^R$) | 78.47 | 64.53 | 0.14 |
| w/o GT-texture resizing | 83.72 | 97.00 | **0.16** |

## 6 Conclusion

We introduced UV3-TeD, a new method for representing and generating textures on sparse unstructured point-clouds constrained to lie on the surface of an input mesh. Our framework is based upon denoising diffusion over surfaces, in which we introduce a new *farthest-sampled multi-headed attention layer* diffusing and capturing features over distant regions, required for coherent texture synthesis. To perform diffusion on meshes of arbitrary topologies, we proposed a mixed Laplacian, fusing both geometric and topological cues. In addition, we proposed online sampling strategies for efficiently working with different quantities related to the shape spectra. Acknowledging that rendering is as equally important as the texture representation, we proposed a path-tracing renderer tailored for our point-cloud textures living on shape surfaces.

**Limitations & Future Work**. Existing UV-based texturing pipelines are heavily engineered, leveraging the recent advances in image generation. We expect that our approach will similarly benefit from the advances in 3D foundation models. Learning high-frequency texture details requires significant training effort, usually exceeding thousands of epochs. More efficient architectures, utilising pooling are required to overcome the drawback and increase the resolution of the generated textures. To enhance quality even further, we recommend extending our method to support BRDFs generation and encourage additional research into sampling strategies capable of ensuring crisp texture borders between parts. With UV3-TeD and its promising results, we invite the community to re-think efficient texture representations, and pave the way to seam-free high-quality coding of appearances on surfaces. As such, we believe our work opens up ample room for future studies in texture generation and other applications requiring the generation of signals that reside on surfaces. For instance, our framework could be easily adapted to applications ranging from HDRI environment map generation, shape matching, and weather forecasting, to molecular shape analysis and generation.

**Broader Impact**. We believe our approach will have a predominantly positive impact, fostering research in generating UV-free textures and ultimately improving creative processes across various industries and empowering individuals with limited artistic skills to participate in content creation. Also, we do not expect nor wish to replace artists due to advancements in the field. Instead, we aim to make their work more efficient, allowing them to unlock their creativity faster.

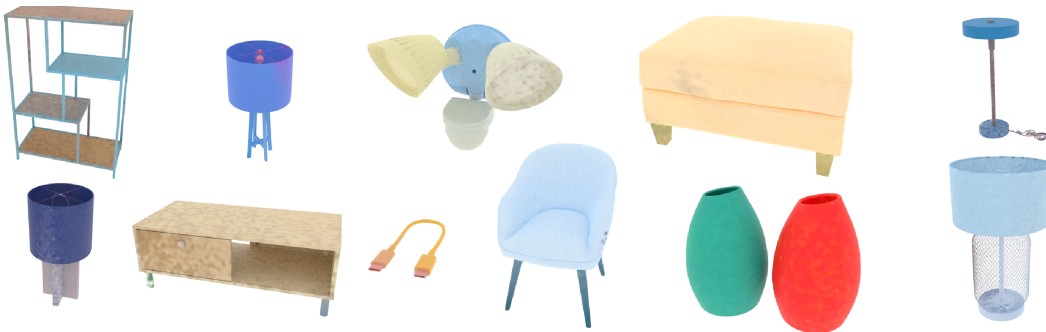

Figure 8: Random samples generated by UV3-TeD (our method) on ABO test set.

## Acknowledgments and Disclosure of Funding

This work was supported by the UK Engineering and Physical Sciences Research Council (EPSRC) Project GNOMON (EP/X011364) for Imperial College London, Department of Computing.

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

# A  Appendix

This document supplements our paper entitled **UV-free Mesh Texture Generation with Denoising and Geodesic Heat Diffusion** by providing further information on our architecture and design choices, additional experiments and ablations for our mesh diffusion framework as well as qualitative results in the form of texture mesh renderings. We also provide the failure cases of UV3-TeD.

## A.1  Additional Information on UV3-TeD

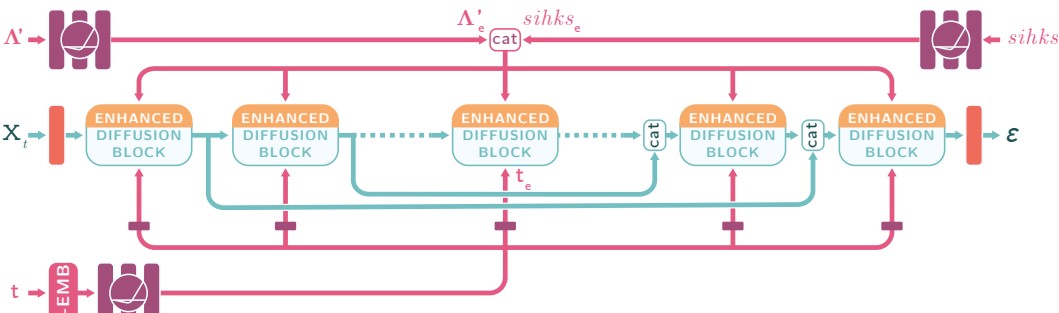

Figure 9: Architecture of the proposed UV3-TeD. The noised input colours ($\mathbf{X}_t$) go through a U-Net-shaped network with multiple Attention-enhanced Heat Diffusion Blocks (detailed in Fig. 4). Blue arrows depict how the blocks are connected in a U-Net fashion. Each Diffusion block is conditioned on a time embedding obtained by embedding the timestep and processing it with an MLP and multiple block-specific linear layers. The attention part of the attention-enhanced blocks is conditioned on the signal obtained by processing $\mathbf{\Lambda}'$ and $sihs$ with two separate MLPs. All the conditioning is depicted in pink arrows.

**Architecture**. As mentioned in Sec. 4, although we do not have pooling and unpooling operators, the skip-connections of our UV3-TeD are inspired by U-Net. Therefore, the output of the first block will be fed to the last block alongside the features coming from the previous layer, the output of the second block will be fed to the penultimate block, et cetera. Given the resemblance to U-Net, we refer to the first half of the network as the downstream branch and to the second half as the upstream one. The block between these two branches is simply referred to as the middle block, which is the only one just receiving a set of features from the previous block and passing its output features to the following block. In our experiments, we use a single middle block as well as 5 blocks in both the down- and up-stream branches (Fig. 9). Each block is built as an attention-enhanced heat diffusion block, which was previously depicted in Fig. 4 and described in Sec. 4.1. Each attention-enhanced diffusion block receives three conditioning signals: a DDPM timestep, a global shape descriptor and a local set of intrinsic coordinates. Each timestep is first converted into a sinusoidal embedding [26], then it goes through a MLP and a block-specific linear layer and it is used to condition the DiffusionNet block part of our enhanced blocks (see Fig. 4). The global shape descriptor conditioning is obtained by processing the normalised and straightened eigenvalues $\mathbf{\Lambda}'$ (Eq. (2)) with a MLP, while the intrinsic coordinate conditioning by passing the scale-invariant heat kernel signatures ($sihks$) through a per-point MLP. These two geometric embeddings are concatenated and passed onto every Diffused Farthest-Sampled Attention layer (Sec. 4.1). Point-wise linear layers are also used as first and last layers of the whole architecture.

The noised input colours ($\mathbf{X}_t$) and the predicted noise $\epsilon = \epsilon_\theta(\mathbf{X}_t, t, \mathbf{\Lambda}', sihks)$ have 3 colour channels each ($R, G, B$). The first linear layer converts the 3 channels into 256 features, while the last one does the opposite. All the MLPs in the attention-enhanced diffusion blocks have ReLU activation functions and 3 layers of size 256. In the upstream branch, the linear layer in the skip connection (Fig. 4) reduces the 512 incoming features to 256. This is not necessary for the downstream blocks as they do not receive additional inputs. The farthest-sampled multi-headed self-attention layers have 8 heads with 64 channels each and operate on 250 farthest samples. The time embedding module produces time embeddings of size 256 and goes through a MLP with SiLU activations and 3 layers of size $[256, 1024, 1024]$. The MLPs for the geometric embeddings also use SiLU activations, but the one processing $\mathbf{\Lambda}'$ has 3 layers of size $[128, 64, 16]$, while the one processing the $sihks$ has 3 layers of size $[32, 64, 16]$.

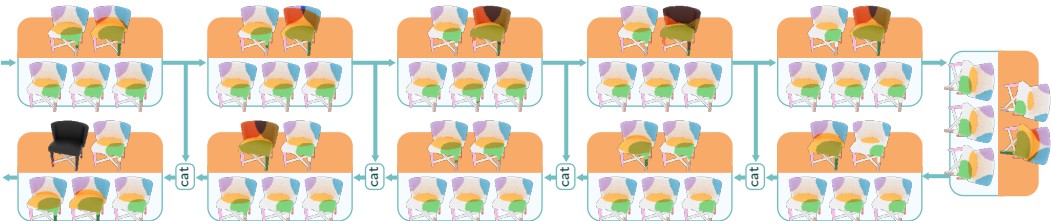

Figure 10: Visual representation of the mean learned diffusion times for the heat diffusion operations in each block of our Network. Conditioning information were omitted for sake of clarity. Refer to Fig. 9 for a more detailed representation of the architecture and to Fig. 4 for the content of each block. The orange part of each block illustrates the diffused farthest-sampled attention layer with its two heat diffusions. The light-blue parts illustrate the three DiffusionNet blocks with their diffusion operations. These mean diffusion times are referred to UV3-TeD trained on ABO and they are displayed by diffusing heat always on the same shape.

**Learned Diffusion Times**. As mentioned in Sec. 3, each heat diffusion operation is performed channel-wise with a learned diffusion time that controls the support of the neural operator. In Fig. 10 we visualise the learned diffusion times of each heat diffusion. In particular, we average the learned time across channels and diffuse heat using the mean time from 8 farthest sampled sources. Interestingly, most Diffusion blocks have a similar support, with the exception of the last two operators, which have a slightly bigger support. In addition, the first diffusion of each diffused farthest-sampled attention layer has a support that is comparable to the Diffusion blocks, while the second diffusion learns longer diffusion times. These longer diffusion times prove the correct operation of the proposed diffused farthest-sampled attention layer (Sec. 4.1) which is supposed to spread the output of the multi-headed self-attention from the farthest samples to all the neighbouring points.

**Fast Gradient Computation**. As briefly mentioned in Sec. 4.1, we propose a fast gradient computation implementation leveraging batched tensor operations. Our efficient implementation, avoiding the usage of multiple nested loops, is more suitable for online computation and does not require to precompute gradients like in the original DiffusionNet [48].

With the objective of constructing a sparse matrix that represents the spatial gradients of the mesh, we start by creating a batched tensor containing information about the neighbours of each vertex and the tangent vectors of the edges connecting them. Since we operate on a point-cloud, the neighbours are estimated with a kNN search with $k = 30$ (like in [48]). Having a fixed number of neighbours facilitates the creation of a batched tensor.

Next, we construct the column and row indices for the sparse matrix. The column indices are created by concatenating an array of vertex indices with the index of their neighbours retrieved from the batched tensor previously computed. The row indices are created by repeating vertex indices $k$ times. We then calculate the values to be stored in the sparse matrix. This involves computing the least squares solution of a linear system between the batched tensor and a matrix constructed by concatenating a column vector of -1's to an identity matrix. The result is split into two parts that are stored as a sparse complex tensor representing the gradients of the mesh, where each row corresponds to a vertex, each column corresponds to a neighbouring vertex, and the value at a specific row and column represents the gradients of the edge connecting the two vertices.

The computational speedup with respect to the original implementation leveraging the multiple nested loops is significant: with approximately $100k$ points our implementation is on average $35\times$ faster, while with approximately $25k$ points it is on average $38\times$ faster.

**Runtimes and Computational Resources**. We run our models on a single Nvidia A100 with 40GB of dedicated memory. The time required to generate a single point-cloud texture ($P^* = 5,000$) is approximately 1 minute and 30 seconds. One training epoch takes approximately 23 minutes with the ABO dataset and 10 minutes with the chairs of ShapeNet. Note that in order to keep a constant GPU utilisation and prevent data loading and processing bottlenecks during online sampling it is advisable to use multiple workers. We use 12 workers on a computer with an AMD EPYC 7763 64-Core Processor, which has a max CPU frequency of 3.5 GHz.

Experiments were conducted using a private infrastructure, which has a carbon efficiency of 0.166 kgCO$_2$eq/kWh. A cumulative of 2, 640 hours (110 GPU-days) of computation was performed on hardware of type A100 PCIe 40GB (TDP of 250W). Total emissions are estimated to be 109.56 kgCO$_2$eq of which 0 percent was directly offset. Estimations were conducted using the Machine-Learning Impact calculator presented in [28] while the carbon efficiency was estimated using the following electricity maps.

## A.2   Additional Experiments

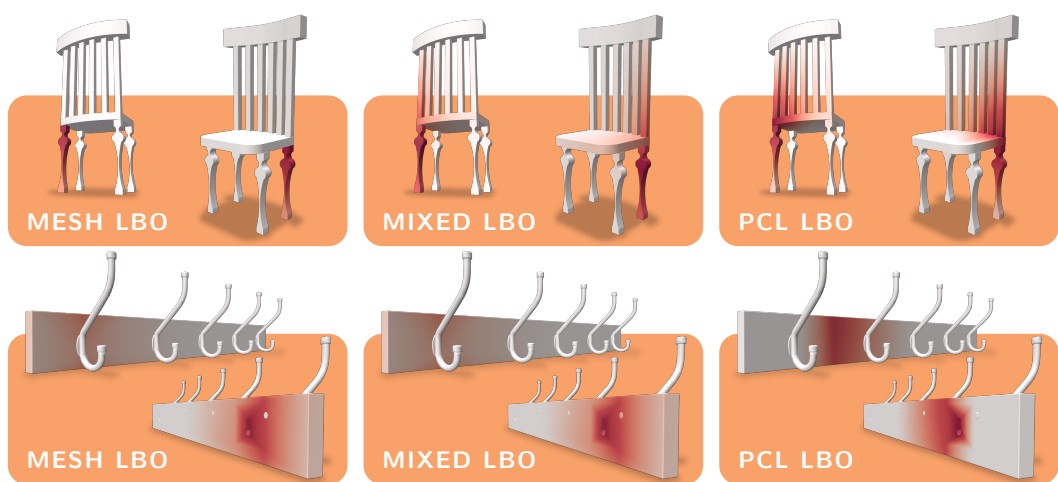

Figure 11: *Top:* heat diffusion on a chair whose legs are disconnected from the rest of the structure. Diffusing heat with the mesh LBO, heat cannot spread beyond the leg where heat was applied. With the point-cloud LBO heat diffuses also across the rest of the structure, but it quickly travels also horizontally across the vertical bars. With our mixed LBO formulation heat correctly spread over the rest of the structure and it appears to better follow geodesic paths. *Bottom:* Heat diffusion on an object with thin structures using different LBO formulations. Using the proposed mixed LBO heat diffuses geodesically, closely mimicking the behaviour of heat diffusion with the mesh LBO. On the contrary, with the point-cloud LBO heat would be immediately transferred from the back of the hanger to its front because of the Euclidean proximity of the two parts. This is an undesirable behaviour not shown by our method.

**More on Heat Diffusion with the Mixed Laplacian**. As detailed in Sec. 4.2, we propose a hybrid formulation of the Laplace Beltrami Operator that we call the Mixed LBO and which is obtained as a convex combination of the mesh and point-cloud LBOs (computed on the vertices of the mesh). As we already explained, by leveraging our LBO we can diffuse heat on surfaces with topological errors and disconnected components (Fig. 5). Fig. 11 (*top*) shows another example where the legs of a chair were not topologically connected to the rest of the structure. Unlike the mesh LBO, our mixed LBO formulation allows heat to spread over the rest of the chair. In addition, when compared to the point-cloud LBO, our formulation retains the topological information provided by the mesh and lets the heat diffuse geodesically. This is particularly evident in Fig. 11 (*bottom*), where some heat is diffused starting from the back-side of a coat hanger. With the mesh LBO heat diffuses mostly on the back of the object, and partially starts to diffuse towards the front. On the contrary, with the point-cloud LBO heat is equally diffused on the front and on the back of the object. Therefore, in the presence of thin structures, it is clear that heat is not geodesically diffused. On the other hand, with our formulation, we avoid spreading heat on the front of the object and we closely mimic the correct behaviour of heat diffusion with the mesh LBO. It can also be noted that with our Mixed LBO some small proportion of heat spreads over a screw placed close to the heat source. We consider this to be an acceptable behaviour. Intuitively, this information-sharing may carry some useful insights on how the screw may be coloured with respect to the surrounding material used to build the hanger. Yet, the amount of heat is lower than in the portion of the hanger touching the screw, facilitating the distinction between different structures.

**Test Online Sampling Strategies**. In Sec. 4.3 we introduced a set of sampling strategies to re-use as much as possible pre-computed operators, make possible the online sampling of point-clouds,

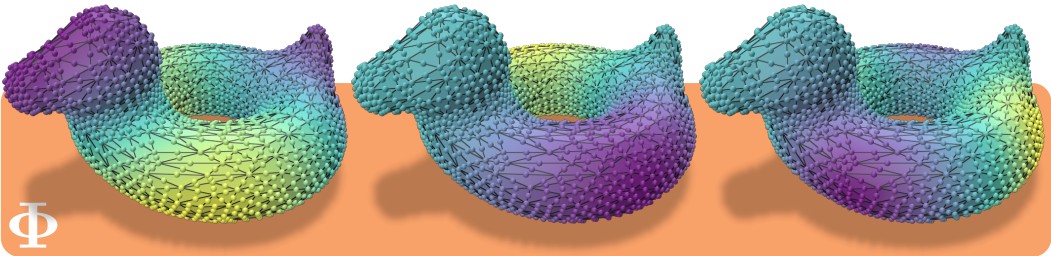

Figure 12: Comparison between the eigenvectors of the same shape with two different sampling densities. The eigenvectors of the original mesh are represented as colours on the surface of the mesh, while the eigenvectors of a mesh obtained subdividing the faces of the original mesh are reported on a point-cloud. The point colours match those of the underlying mesh, suggesting that sampling the eigenvectors of the mesh at the point locations would produce the same result.

and ensure that heat can be diffused following the known surface information of the mesh. After computing and eigendecomposing our Mixed LBO, we store its eigenvectors, eigenvalues, and mass matrix. The eigenvalues remain unchanged when sampling and therefore are re-used without any modification. The mass matrix, being proportional to the distance between the sampled points, can be approximated as described in Sec. 4.3 from the PDS radius. Finally, the eigenvectors are recomputed by interpolating their values on the vertices of the mesh. In Fig. 12, we empirically show that this interpolation operation is possible and produces the same results as recomputing the eigenvectors. In particular, after computing the eigenvectors of a mesh, we subdivide it and compute the eigenvectors of the subdivided mesh, which are displayed as a coloured point-cloud. As it can be observed in Fig. 12, the values of the coloured point-cloud are in agreement with the colours on the surface of the mesh, which are obtained as a linear interpolation between the vertex colours. For this reason, it is not necessary to re-compute the eigenvectors for every point-cloud that we sample and we can interpolate the mesh values instead.

Established that we can interpolate the eigenvectors, we still need to prove that diffusing heat on a point-cloud sampled with PDS, whose eigenvectors have been interpolated and the mass has been approximated from the PDS radius, produces the same results as diffusing heat on the surface of the mesh. Therefore, we compare the heat diffused –from the same source– on a mesh and on an online-sampled point-cloud. As we can observe from Fig. 13, the two diffusion processes produce the same results.

**Additional Samples**. More textures generated on test shapes by our method are reported in Fig. 14 and Fig. 15 *top*. Fig. 16 shows more textures generated by UV3-TeD on ABO objects and rendered from multiple viewpoints. This proves how, unlike methods relying on multi-view images for texture generation, UV3-TeD generates textures directly on the surface of the objects, making them multi-view consistent by construction. We also report more chairs generated by Point-UV Diffusion [58] and rendered with our rendering method (Sec. 4.4) for fairness of comparison. Finally, Fig. 19 reports some failure cases of our method.

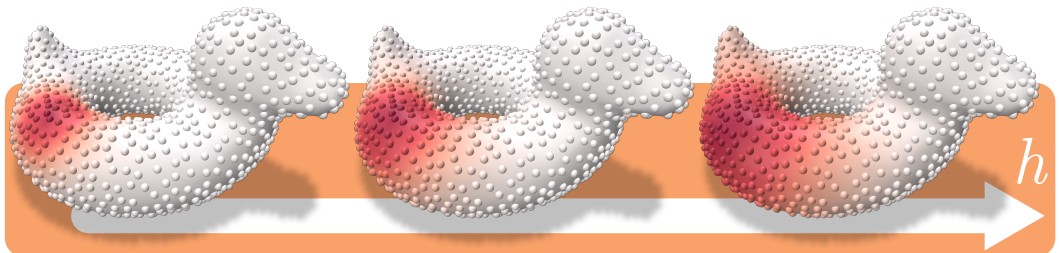

Figure 13: Heat diffusion computed on the vertices of a mesh and with our online sampling on a point-cloud. The heat depicted on the surface of the mesh is computed with the traditional method using Eq. (1) and our Mixed LBO (Sec. 4.2). The heat depicted on the point-cloud was computed using the online sampled operators like described in Sec. 4.3. Heat diffuses in the same way with both techniques proving the correctness of our sampling strategy.

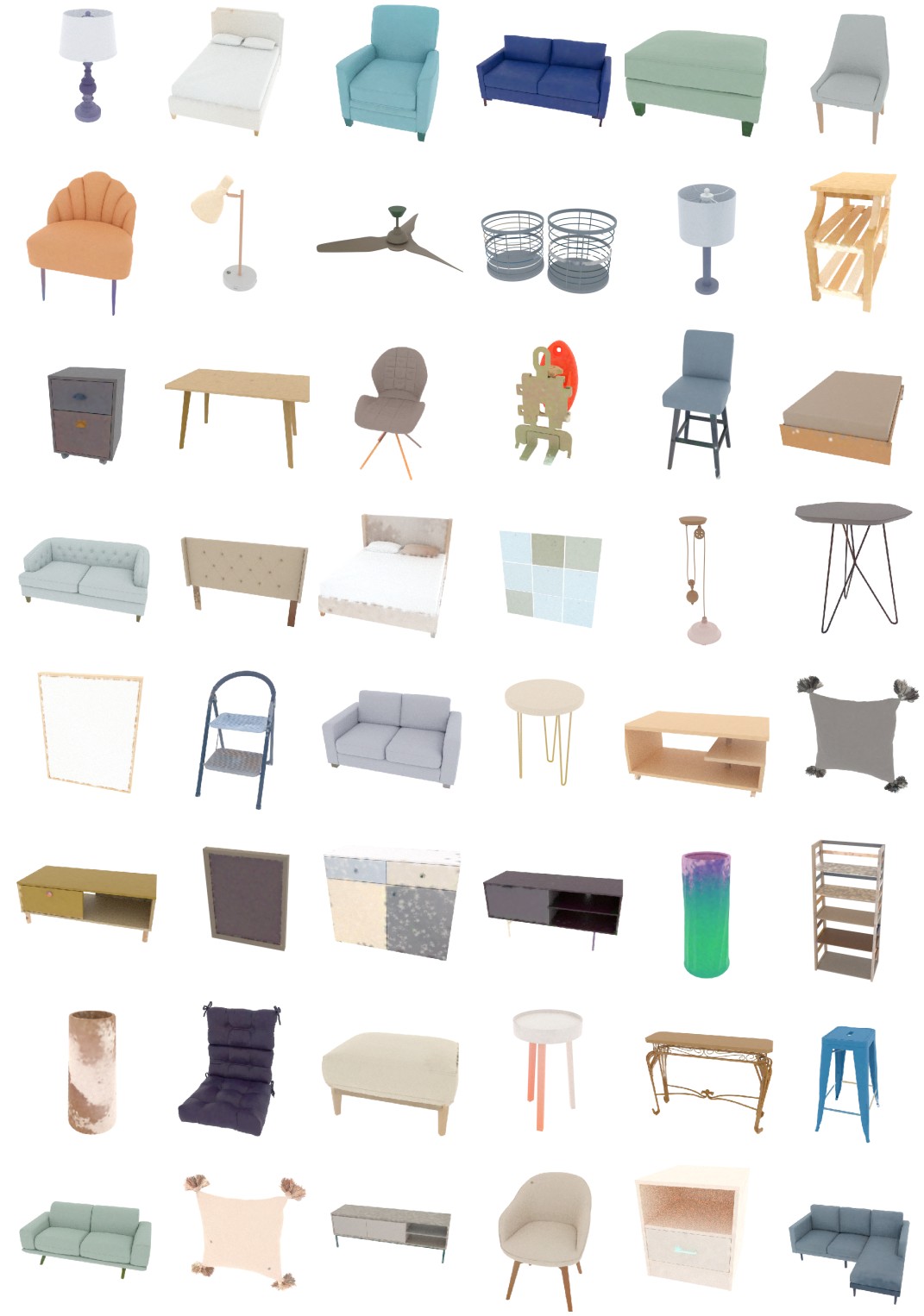

Figure 14: Additional random samples generated by our UV3-TeD trained on ABO.

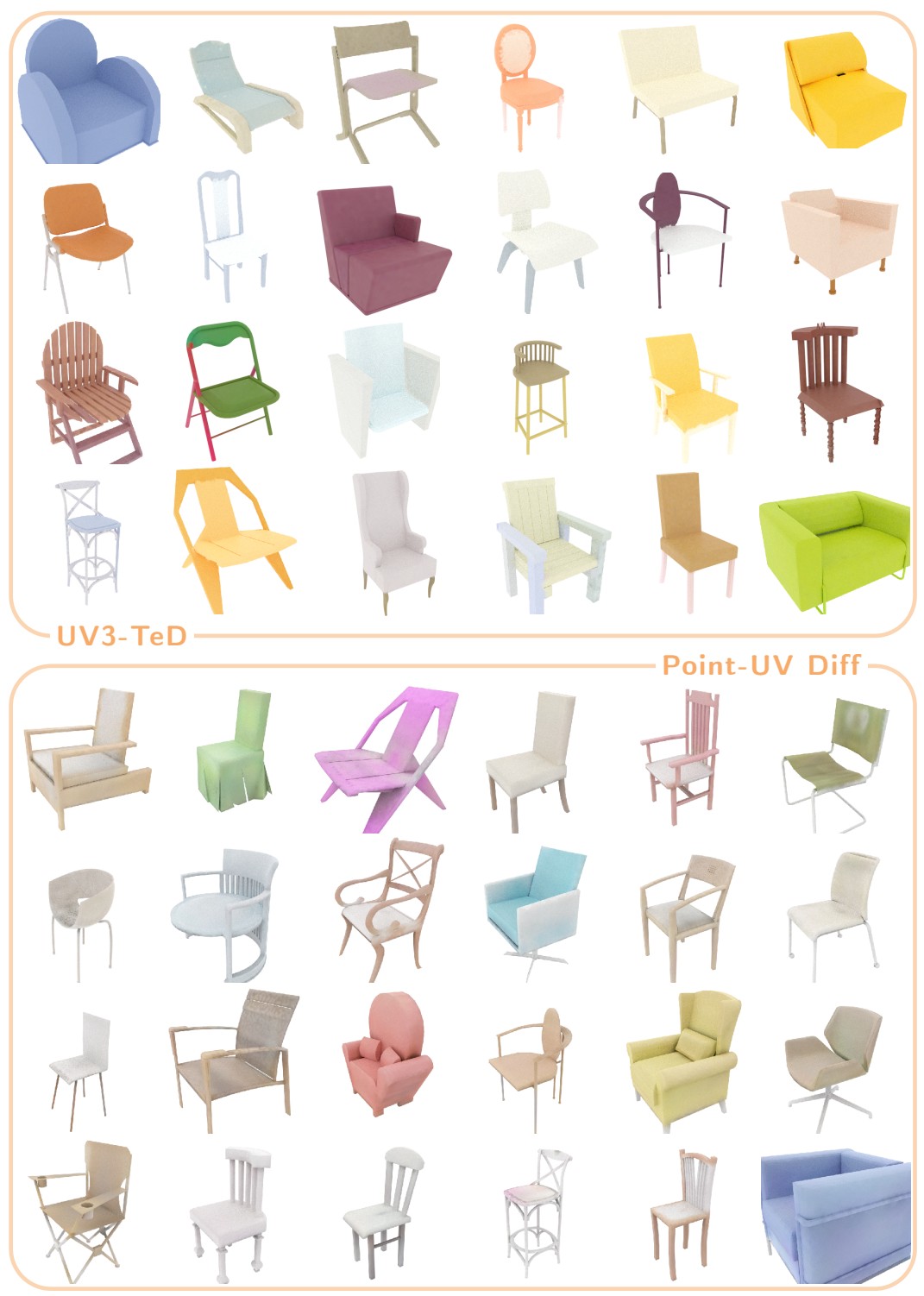

Figure 15: Additional random samples generated by our UV3-TeD (*top*), and Point-UV Diffusion (*bottom*) trained on chairs from ShapeNet. The point-clouds generated by Point-UV Diffusion were rendered with our method for a more fair comparison with no projection artefacts. Our method generates more diverse textures and better distinguishes the different parts.

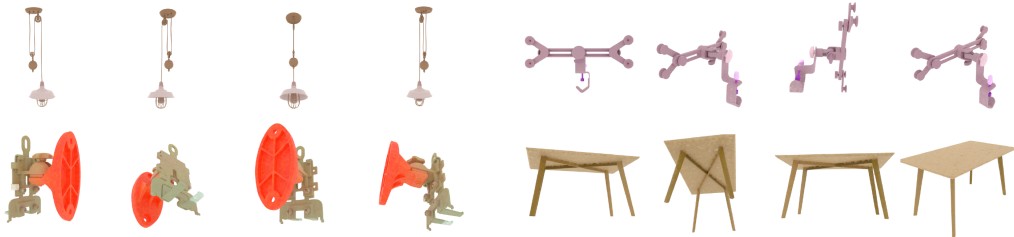

Figure 16: Multi-view consistency of ABO shapes textured with UV3-TeD

**Higher Frequency Point-cloud Textures on CelebA.** The low-frequency nature of the results on ShapeNet chairs and ABO mostly stem from the datasets used. In fact, most objects in ShapeNet have plain per-part colours and can be considered mostly low-frequency textures. ABO on the other hand has more detailed textures, but these textures are much higher in frequency than our sampling resolution (e.g. wood grain, rubber pours, etc.). To demonstrate the ability of our method to handle more complex high-frequency details, we have trained UV3-TeD on CelebA images [29] projected on a plane deformed by a 3D ripple effect. UV3-TeD was trained for 50 epochs, with a learning rate of $5e^{-4}$, and a batch size of 4. The farthest point samples were reduced to

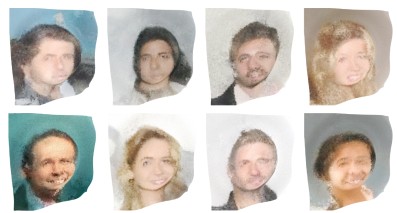

Figure 17: Plane perturbed by a ripple effect and textured with point-cloud textures generated by UV3-TeD trained on CelebA for only 50 epochs.

$S = 200$ and the channels in the farthest-sampled self-attention layers to 64. The target PDS samples ($P^* = 5,000$) as well as all the other hyperparameters were the same as those previously detailed in the Implementation Details (Sec. 5) and Architecture (App. A.1) paragraphs. This experiment shows promising results (Fig. 17) considering that it is capable of generating diverse CelebA textures even prior to reaching full convergence.

Considering that our Diffusion Blocks resemble the original operators of [48], we believe our results also prove the unproven claim of heat-diffusion-based operators being capable of operating at high-frequencies.

**Ablation Studies Detailed Discussion**. In Sec. 5 we performed multiple ablations studies to examine how much each model component, conditioning, and choice contributes to the overall performance. Ablations were quantitatively evaluated using three main metrics: the FID, KID, and LPIPS scores. The first two evaluate the visual quality of generated samples rendered from random viewpoints, while LPIPS evaluates the perceptual dissimilarity between different objects consistently rendered from the same viewpoint. Results were reported in Tab. 2.

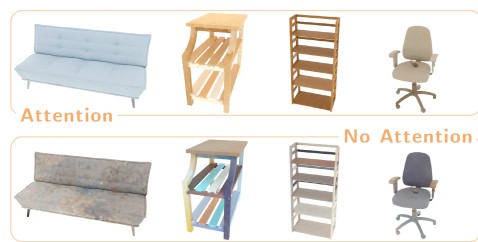

Figure 18: ABO shapes textured by UV3-TeD with and without the proposed farthest-sampled attention layer.

We started by investigating the effects of the proposed farthest-sampled attention layers, removing the proposed attention causes the most significant drop in performance across metrics, showing the positive impact of our proposed attention mechanism. The benefits provided by the attention mechanism can be observed also in Fig. 18. It is clear that this mechanism makes the generated textures more realistic and uniform across different parts.

We then proceeded to remove the normalised and straightened eigenvalue ($\mathbf{\Lambda}'$) conditioning as well as the scale-invariant heat kernel signatures ($sihks$) conditioning. Both cause a drop in visual quality. However, the absence of $\mathbf{\Lambda}'$, which holds class-related information, appears to slightly improve the diversity of the generated point-cloud textures at the expense of sample quality. We believe that this result still signals the importance of providing both conditioning signals. Also directly using the online-sampled eigenvectors $\mathbf{\Phi}_p$ instead of $sihks$ causes a small drop in performance. Note that

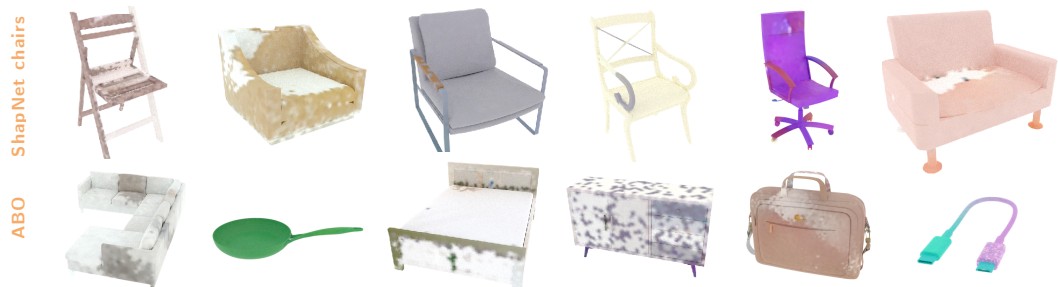

Figure 19: Failure cases of random samples generated by our UV3-TeD trained on ShapeNet chairs (*top*) and ABO (*bottom*). Most failures exhibit issues in correctly recognising the different object parts, long-range inconsistencies, uniform yet unreasonable colours, or blotchy patterns.

the diffused farthest-sampled attention layers contain a group normalisation layer after the first heat diffusion. Since it expects a specific number of channels, when either $sihks$ or $\mathbf{\Lambda}'$ are removed, to make the ablation more controllable, instead of modifying the group normalisation layer we duplicate the remaining conditioning thus providing redundant information.

Although we have extensively proved the validity and the advantages of our mixed LBO operator (e.g., Fig. 5 and Fig. 11), we compare the performance of our network with the model trained to compute heat diffusions with our online sampling and using the mesh ($\mathbf{L}_m^R$) or point-cloud ($\mathbf{L}_p^R$) LBOs. Our method performs slightly worse in terms of LPIPS when compared to using the mesh Laplacian, but it also exhibits a far greater performance improvement in terms of FID and KID. This, and the previous geometrical evaluations of the mixed LBO, make the adoption of our mixed operator preferable. Similarly, we can reach the same conclusion by observing the results with the point-cloud operator, which is also not improving over the final UV3-TeD model in terms of LPIPS score. It is worth noting that ABO is a well curated dataset with a reduced number of topological defects. Therefore, we expect performance to deteriorate even further on less curated datasets.

Finally, we evaluate the effects of our texture resizing Sec. 4.3. Although the LPIPS score improves, showing an increased diversity of the generated samples, the blotchy patches depicted in some of the failure cases on Fig. 19 become present on most generated point-cloud textures. This results in worse FID and KID scores when compared to our full model.

As already mentioned in Sec. 5, our final model (UV3-TeD) reports the best quality scores while maintaining good diversity scores.

**A Note on Photorealism.** All the result images of this paper except Fig. 1 were rendered with a constant emitter to better showcase the generated albedos. Nevertheless, our generated albedo textures are intrinsically relightable and can be photo-realistically rendered. More photorealistic renderings can be obtained by rendering objects with environment maps and training our model to generate full BRDFs. Fig. 20 shows how just using an environment map can improve the realism of one of our chairs.

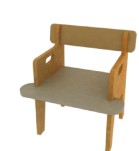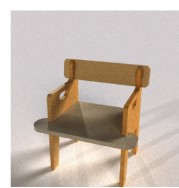

Figure 20: Chair rendered with a constant emitter (*left*) and an environment map (*right*).

