# OpenReview forum: "UV-free Texture Generation with Denoising and Geodesic Heat Diffusion"
_NeurIPS.cc/2024/Conference — NeurIPS 2024 poster_

### Official Review · Reviewer_CcU9 · 2024-07-01

**Soundness:** 2
**Presentation:** 3
**Contribution:** 2
**Rating:** 5
**Confidence:** 4

**Summary:**

The paper proposed a new way to represent the texture to replace the UV-map and trained a model for texturing generation of objects that are not limited on a specific category.

**Strengths:**

The proposed new representation avoid per-category training and seams on UV maps.

**Weaknesses:**

1. Missing the experiments on the Objaverse dataset.
2. The Figure 3 is not detailed enough to describe how the entire pipeline work.

**Questions:**

1. Any intuitive explanation of what the "h" is?
2. Is it possible to extend the proposed method to text-conditioned texture generation?
3. Can this method generate textures with diverse textures like objects in Objaverse?

**Limitations:**

1. It seams that the generated textures are almost in pure colour or with limited details. I am really concerned about this point. There has already been many texture generation methods that can produce diverse, detailed and photo-realistic textures.

---

> ### Author Rebuttal · Authors · 2024-08-07
>
> We thank the reviewer for their time and efforts invested in our paper. Our detailed responses are given below.
>
> > Fig. 3 is not detailed enough to describe how the entire pipeline work.
>
> We thank the reviewer for suggesting us to include the full pipeline figure, which we provided in the supplementary attached PDF (Fig. A). We have also included this in our revision, nicely complementing Fig. 8 in section A.1 depicting our architecture and Fig. 3 showing its core building block. We hope that this eases the understanding of our method.
>
> > It seams that the generated textures are almost in pure colour or with limited details. I am really concerned about this point. There has already been many texture generation methods that can produce diverse, detailed and photo-realistic textures.
>
> Although we acknowledge that many state of the art methods operating in UV-space (such as those mentioned in Sec. 1, line 31) have generated impressive results, we want to highlight that they operate on images, a data type which has been vastly explored by the Deep Learning community and where models are well-engineered, mature in terms of efficiency and quality, and trained on larger datasets.
> We find it important to note that we do not aim to compete against UV techniques, but rather attempt to prompt a paradigm shift towards a direction that will not require UV-mapping, with its many unsolved issues.
> The best non-UV-based method currently available is arguably the coarse stage of Point-UV diffusion. As can be observed in Fig. 6 and Fig. 14, our method generates more diverse and realistic textures that better respect the semantics of different objects' parts. In addition, it is worth noting that most objects in ShapeNet and ABO do have limited texture details at a resolution that could be captured by current non-UV models like ours. For these reasons, we consider our results to be an important, promising step towards the generation of UV-free textures and we are confident that improving the efficiency of our architecture with the use of pooling (line 343) will enable the generation of high-resolution point cloud textures.
>
> Also, unlike many other texture generation methods, we generate textures that do not factor in the environment. Our renderings were performed with a constant emitter to better showcase the generated albedo textures. Nevertheless, photorealism can be significantly improved rendering the textured objects with environment maps (see Fig. B1 in the attached PDF).
>
> Finally, to demonstrate the ability of our method to handle more complex high-frequency details, we have trained UV3-TeD on CelebA images projected on a plane deformed by a ripple effect. This experiment showed promising results considering that it is capable of generating diverse CelebA textures even without reaching full convergence (see Fig. D in the attached PDF). We have added this experiment and its detailed description in the appendix.
>
> > Any intuitive explanation of what the "h" is?
>
> As detailed in lines 162-164: "$h$ is the channel-specific learnable parameter indicating the heat diffusion time and effectively regulating the spatial support of the operator, which can range from local ($h \rightarrow 0$) to global ($h \rightarrow +\infty$)." In other words, it is the heat diffusion time.
>
> > Is it possible to extend the proposed method to text-conditioned texture generation?
>
> Although this is not something we have experimented with, we believe it is possible to extend UV3-TeD to support text conditioning.
> We believe there are two potential approaches to providing text conditioning, each with its challenges. The first approach would require re-training the model using existing 3D datasets with text descriptions and concatenating CLIP embeddings of texture descriptions to the shape conditioning vectors. Since the captions in these datasets usually combine shape and texture information, the main challenge is the potential need to remove global shape information from the ground truth textual descriptions to prevent any clash with our global shape descriptors, which proved to be quite effective (Tab.2). The second and more promising approach would be to use UV3-TeD as a prior and run an optimisation according to an auxiliary differentiable constraint like in [a]. The constraint could be implemented with a CLIP loss between the desired text prompt and the rendered image of the textured object. The mild challenge here is in making our rendering pipeline fully differentiable.
>
> > Can this method generate textures with diverse textures like objects in Objaverse?
>
> We believe it is possible, but it is not trivial. Only a subset of data would be suitable for training. Data with no texture, low quality scans, and scenes with multiple objects (e.g., entire houses with all the furniture inside) need to be removed. Successfully automating this process is essential yet extremely complex. More importantly, the current training procedure is prohibitively slow given our computational resources. Training on approximately 200k shapes (corresponding to 1/4 of Objaverse) takes approximately 12h per epoch. As mentioned in the limitations and future works section, we believe that ''more efficient architectures, utilising pooling are required to overcome the drawback" (line 342-343). Given our current computational limitations, we have conducted additional experiments aimed at showcasing the model's ability to work with higher frequencies on CelebA (see above).
>
>
>
> [a] Graikos et al. Diffusion models as plug-and-play priors. NeurIPS2022.

---

> > ### Comment · Reviewer_CcU9 · 2024-08-12
> > **Response to the rebuttal**
> >
> > I am satisfied with the rebuttal and have lifted up my rate. Good luck!

---

> > > ### Author Response · Authors · 2024-08-12
> > >
> > > We appreciate that the reviewer has considered our feedback and reflected this in their final rating. We have incorporated all of our responses and discussions into the revision.

---

### Official Review · Reviewer_cPWU · 2024-07-13

**Soundness:** 2
**Presentation:** 3
**Contribution:** 3
**Rating:** 6
**Confidence:** 3

**Summary:**

This paper proposes to apply denoising diffusion generative model directly on mesh surfaces, with a focus on texture generation.
It achieves this goal by utilizing the DiffusionNet method which utilizes heat diffusion to enable 'convolution' and message passing within the surface.
This work modify and extend the original DiffusionNet mainly in the following way:

**Mixing the extrinsic and intrinsic adjacency**.
DiffusionNet uses pure intrinsic connectivity (Mesh LBO) by default, where the information only flow within the surface without considering the actual adjacency in 3D space.
This may introduce problems for apply DDPM on arbitrary meshes. Instead, the proposed method mixes intrinsic adjacency with extrinsic adjacency (Point cloud LBO) so that touching parts will also have mutual information flow.

**Adding attention mechanism to facilitate long-range consistency**
Since heat diffusion typically only disperse or aggregate information locally, this paper adds an attention mechanism to better model dependencies that are far apart.

**Strengths:**

**Denosing Diffusion on Manifold via Heat Diffusion**.
Although the two diffusion mechanism have very different nature, it is very interesting to see these two diffusion can work together to enable direct generative modeling on manifold. I believe this paradigm **has greater potentials** than just mesh texturing. For example, one can easily adopt this framework for generating spherical or toric images (for example, generate HDRI environment map) or any signal that resides on geometric structures that are more complex than regular 2D grid.

**Good paper exposition**.
The paper presentation is clear and the paper figures explain the concepts in an easy-to-follow manner.

**Weaknesses:**

**No much explanations on the results / comparisons**

**The neccessity of texturing the points using an intrinsic (hybrid) architecture**

Please see the questions section to see further details.

**Questions:**

What is the DiffusionNet DDPM baseline?
It seems the paper does not include a description for this baseline.

What is exactly the benefit of using diffusion on the surfaces vs in the embedded 3D space?
It seems the results are better, but why?
Why providing the manifold neibourhood information neccessarily facilitate the texture generation?
How do you compare by simply using a fully transformer architecture? It can directly model long range dependency quite well, without the need for LBO. The proposed method has an emphasis on incorporating attention into the DiffusionNet framework. But why not directly use the Transformer that are full of attention blocks?

Maybe transformation invariance is one advantage of intrinsic message passing, but one can always enhance this by using data augmentation of random transformation.

I would suggest to **show some qualitative analysis examples** that conveys intrinsic message passing indeed is vital in some cases for colorizing surface points.

**Limitations:**

The authors discuss the limitation of generating textures with high-frequency.

---

> ### Author Rebuttal · Authors · 2024-08-07
>
> We thank the reviewer for their positive and encouraging feedback. We are glad the reviewer believes our work has greater potential than just mesh texturing. We agree that our approach can easily adapt to other applications requiring the generation of signals that reside on geometric structures that are more complex than regular 2D grids. We will mention the suggested applications as potential future work. We also address the major questions below.
>
> > What is the DiffusionNet DDPM baseline? It seems the paper does not include a description for this baseline.
>
> We thank the reviewer for bringing this to our attention. We have now added more details on this baseline. The DiffusionNet DDPM was built as a DDPM model with a U-Net-like architecture with as many layers as ours, but without shape conditioning and using the Vanilla DiffusionNet blocks introduced in [54]. Therefore, not only there were no farthest-sampled attention layers, but no online sampling strategy was used and the point-cloud Laplacian and mass matrix were used. Colours were still sampled like for UV3-TeD. All the hyperparameters matched ours.
>
> > What is exactly the benefit of using diffusion on the surfaces vs in the embedded 3D space? It seems the results are better, but why? Why providing the manifold neibourhood information neccessarily facilitate the texture generation?
>
> The main benefit of using heat diffusion on the surface is enabling geodesic information sharing by construction.
> By looking at Fig. 10 in Appendix A2 we can observe how heat diffusion performed with a point cloud Laplacian struggles to correctly aggregate information. Note that point cloud Laplacians are built by creating a new triangulation that attempts to capture the surface properties of the shape on which they are built. Therefore, when they are used in heat diffusion we can expect the heat to spread quasi-geodesically. Performing diffusion operations directly in the embedded 3D space would then further exacerbate the issues showcased in Fig. 10 and described in Appendix A2.
>
> > How do you compare by simply using a fully transformer architecture? It can directly model long range dependency quite well, without the need for LBO. The proposed method has an emphasis on incorporating attention into the DiffusionNet framework. But why not directly use the Transformer that are full of attention blocks?
>
> Global information sharing provided by transformers is indeed powerful, particularly for capturing long-range dependencies. However, for tasks like texture generation, which resemble image generation, local information sharing is equally crucial. Our approach integrates both attention mechanisms and convolution-like operations, leveraging the strengths of both. Mixing attention and convolutions is a standard approach also in many state-of-the-art diffusion models operating on images, providing a balance between capturing global context and preserving local details while keeping low computational and memory overhead.
>
> Moreover, unlike images, there is no de-facto standard to choosing a 3D representation. In a variety of cases, ignoring the 2-manifold nature of 3D objects yields inferior results. Hence, any transformer of choice should be adapted to operate in restriction to the object surface, requiring significant engineering efforts (see for instance [i]).
> Our approach provides a natural way to achieve this. Our novel Laplacian also enables us to span a range of representations from point clouds to pure mesh topology.
>
>
>
>
>
>  > Maybe transformation invariance is one advantage of intrinsic message passing, but one can always enhance this by using data augmentation of random transformation.
>
> Data augmentation is certainly a strong alternative to transformation invariance, which is indeed a desirable by-product of our intrinsic message passing. This is potentially useful in a variety of applications like texturing digital humans. Nonetheless, our primary motivation for choosing heat diffusion lies not in its invariance properties, but in its ability to effectively propagate information across the object surface.
>
> [i] Chandranet al. Shape Transformers: Topology‐Independent 3D Shape Models Using Transformers. Computer Graphics Forum 2022

---

### Official Review · Reviewer_Yhxo · 2024-07-14

**Soundness:** 3
**Presentation:** 3
**Contribution:** 3
**Rating:** 5
**Confidence:** 3

**Summary:**

This paper proposes UV3-TeD, a 3D mesh texturing method without explicit UV mapping. To circumvent the challenges of using explicit UV mapping, the authors propose to represent texture as a point cloud with color features. While there were some methods that used similar representation, the authors emphasize that their representation can effectively constrain the point clouds to operate only on the mesh surface. To generate the colored point cloud in a generative manner, the authors propose a point cloud DDPM. When designing the diffusion model, the authors design a heat diffusion block augmented with the farthest sampled attention mechanism. The authors evaluate their method on 3D shape datasets such as ShapeNet-Chairs and ABO.

**Strengths:**

- The paper proposes an interesting approach for texturizing 3D meshes. Although the concept of colored point clouds has been vastly explored, the proposed heat-diffusion-based DDPM seems to be effective in constraining the points on the 3D mesh surface.

- The paper presents nice and effective visualizations of the contributions and effects of their proposed methods.

- The idea of farthest-sampled attention seems to be valid and fresh.

**Weaknesses:**

- The visualized results for the generation results seem to be too low-frequency. The concept of heat diffusion may guide the generated textures to be coherent and locally smooth, but it seems to be inherently limited in representing high-frequency details, such as stripes or repeated patterns.

- While the authors proposed farthest-sampled attention to enhance long-range texture smoothness, this reviewer could not find any results that show multi-view or long-range consistent generation results. It'd be more convincing if the authors could include multi-view rendered results of a single mesh, with or without farthest-sampled attention.

- The authors mainly compared their method with PointUV-Diff. However, to show the benefit of using colored point clouds rather than explicit UV mapping or texture maps, it is highly recommended to show some comparison results (both quantitative & qualitative) with existing UV-mapping-based texture generation methods.

**Questions:**

- How does UV3-TeD perform when applied to meshes with highly-detailed surfaces or meshes that require high-frequency textures, e.g., 3D clothed human scans?

**Limitations:**

- The texture generation results seem to be more cartoonish or blurred rather than photorealistic, while it is trained with some near-real datasets, e.g., ShapeNet or ABO.

---

> ### Author Rebuttal · Authors · 2024-08-07
>
> We thank the reviewer for their constructive and interesting comments. We start by addressing two related points and then proceed to provide a detailed response to the remaining comments.
>
> > The visualized results for the generation results seem to be too low-frequency. The concept of heat diffusion may guide the generated textures to be coherent and locally smooth, but it seems to be inherently limited in representing high-frequency details, such as stripes or repeated patterns.
>
> > How does UV3-TeD perform when applied to meshes with highly-detailed surfaces or meshes that require high-frequency textures, e.g., 3D clothed human scans?
>
> We indeed agree that there is a trade-off between high frequency details and local coherence. However, the impact is not as dramatic. In fact, the low-frequency nature of our results mostly stem from the datasets used. In fact, most objects in ShapeNet have plain per-part colours and can be considered mostly low-frequency textures. ABO on the other hand has more detailed textures, but these textures are much higher in frequency than our sampling resolution (e.g. wood grain, rubber pours, etc.). Note that the as discussed in the \textit{Remarks} paragraph of Sec 3.3 in [54], the concept of heat diffusion is used for communication across points while the MLP and the gradient features enable learning high-frequency outputs. Since our Diffusion Blocks resemble the original operators of [54] we expect them to be capable of producing high-frequency results. While [54] never proved this claim, we have now conducted a texture generation experiment by training UV3-TeD on CelebA images projected on a plane deformed by a 3D ripple effect. This experiment showed promising results demonstrating that our model is capable of generating diverse CelebA textures even without reaching full convergence (see Fig. D in the attached rebuttal PDF). We have added this experiment and its detailed description in the appendix.
>
> > While the authors proposed farthest-sampled attention to enhance long-range texture smoothness, this reviewer could not find any results that show multi-view or long-range consistent generation results. It'd be more convincing if the authors could include multi-view rendered results of a single mesh, with or without farthest-sampled attention.
>
> We thank the reviewer for pointing this out. In the attached PDF we have provided additional figures to support this. In particular, we would like to emphasise that unlike methods relying on multi-view images for texture generation, our method generates textures directly on the surface of the objects,  making it multi-view consistent by construction (Fig. C). The benefits provided by the attention mechanism can be observed from Fig. B2 in the PDF. It is clear that this mechanism makes the generated textures more realistic and uniform across different parts. We have added both figures in the Appendix of our revision.
>
> > The authors mainly compared their method with PointUV-Diff. However, to show the benefit of using colored point clouds rather than explicit UV mapping or texture maps, it is highly recommended to show some comparison results (both quantitative \& qualitative) with existing UV-mapping-based texture generation methods.
>
> We thank the reviewer for pointing this out. It is worth noting that Point-UV Diffusion already performed a comparison against Texture Fields and Texturify on the chairs of ShapeNet. Not only their full pipeline (see Tab. 1 of the Point-UV Diffusion paper [54]), but also their coarse stage (see Ablation metrics in Tab. 2 of [54]) outperformed the competing methods in terms of KID and FID. Although neither of these methods operate in UV space, Texturify made comparisons against UV (LTG and UV Baseline), Implicit (Texture Fields), Tri-plane (EG3D), and sparse grid (SPSG) methods, outperforming all these methods across all metrics on the chairs of ShapeNet (see Tab. 1 in Texturify paper). Our method, outperforming Point-UV Diffusion, is expected to surpass all these baselines as well. We have added this discussion to our paper.
>
>
>
> > The texture generation results seem to be more cartoonish or blurred rather than photorealistic, while it is trained with some near-real datasets, e.g., ShapeNet or ABO.
>
> This is a good observation. Besides the quality of the textures in the datasets, our method, albeit being agnostic to the sampling strategy and resolution, is affected by these choices, especially in terms of photorealism and when it comes to generating crisp borders between parts.
>
> More importantly, we generate only albedo textures and render them with a constant emitter to better showcase the generated albedos. More photorealistic renderings can be obtained by rendering objects with environment maps and training our model to generate full BRDFs. Note that this is also a key difference with many texture generation methods like Texturify where the textures are not albedos but rather bake-in the contributions of the self-shadows as well as the light interactions between the BRDF properties and environment maps. We have attached a rendering (Fig. B1) showing how just using an environment map can improve the realism of one of our chairs. We thank the reviewer for bringing this up and now mention this key intrinsic advantage of our method in the revision.

---

### Author Rebuttal · Authors · 2024-08-07

We sincerely thank all the reviewers for their constructive feedback. We appreciate that reviewers find our work interesting, valid, and fresh (Yhxo), with greater potential than just mesh texturing (cPWU), clearly presented (Yhxo, cPWU), and with nice and effective visualisations (Yhxo, cPWU). We fully acknowledge the reviewers’ concerns regarding the need for more compelling experiments. In the individual responses, we have provided robust evidence and explanations, addressing all the major issues raised thoroughly.

The Rebuttal PDF here attached complements the individual responses by providing a framework figure, more photorealistic renderings, additional evidence of our intrinsic multi-view consistency and the benefits of our proposed attention, and the results of high-frequency CelebA texture generation.

We trust that the supplementary clarifications and data we have provided effectively address all the concerns and queries raised by the reviewers.

---

> ### Author Response · Authors · 2024-08-12
>
> Dear Reviewers,
>
> Thank you once again for your time and valuable feedback on our submission. As the author-reviewer discussion progresses, we would greatly appreciate any further updates or insights you may have regarding our rebuttal. If you require any additional clarification, please feel free to let us know.
>
> All the best,
>
> The Authors

---

> > ### Comment · Reviewer_cPWU · 2024-08-12
> > **Thank you**
> >
> > Thanks for the responses. I have no further questions and still support the acceptance of this paper.
> > Also, hope in the future we can see more works of generative modeling on manifold domain that shows more compelling examples on the advantages of geodesic message passing.

---

> > > ### Author Response · Authors · 2024-08-13
> > >
> > > We appreciate the reviewer's continued support for the acceptance of our paper and confirm that we have addressed their comments in the revision. We also agree that generative modeling on surfaces is a promising area of research and hope that our work can inspire more researchers to explore and further innovate geodesic methods.

---

### Decision · Program_Chairs · 2024-09-25

**Decision:**

Accept (poster)

**Comment:**

The paper introduces a novel approach for 3D texture generation by avoiding traditional UV mapping and instead using colored point clouds constrained to mesh surfaces, combined with the DDPM and heat diffusion. This method addresses common UV-based texturing issues like seams and varying resolution. The reviewers appreciated the innovation, particularly the integration of a heat-diffusion-based self-attention mechanism for ensuring spatial consistency, and noted the potential for broader applications beyond mesh texturing.

However, there were concerns about the method's ability to generate high-frequency details, with some results appearing cartoonish or blurred. The evaluation could be more compelling with additional comparisons to state-of-the-art UV-based methods. The authors effectively addressed these concerns during the rebuttal, providing additional experiments and clarifications that satisfied most reviewers, leading to an improved score from one reviewer.

Given the paper's innovative contributions and the authors' engagement in addressing feedback, the paper is recommended for acceptance as a poster presentation. This format will allow the authors to showcase their work while gathering further insights for future improvements.